# Evaluating the Physicochemical Properties–Activity Relationship and Discovering New 1,2-Dihydropyridine Derivatives as Promising Inhibitors for PIM1-Kinase: Evidence from Principal Component Analysis, Molecular Docking, and Molecular Dynamics Studies

**DOI:** 10.3390/ph17070880

**Published:** 2024-07-03

**Authors:** Hanna Dib, Mahmoud Abu-Samha, Khaled Younes, Mohamed A. O. Abdelfattah

**Affiliations:** College of Engineering and Technology, American University of the Middle East, Egaila 54200, Kuwait; mahmoud.abusamha@aum.edu.kw (M.A.-S.); mohamed.abdelmoety@aum.edu.kw (M.A.O.A.)

**Keywords:** PCA, 1,2-dihydropyridine, molecular docking, anti-colorectal tumor activity, molecular dynamics, database mining

## Abstract

In this study, we evaluated the physicochemical properties related to the previously reported anticancer activity of a dataset comprising thirty 1,2-dihydropyridine derivatives. We utilized Principal Component Analysis (PCA) to identify the most significant influencing factors. The PCA analysis showed that the first two principal components accounted for 59.91% of the total variance, indicating a strong correlation between the molecules and specific descriptors. Among the 239 descriptors analyzed, 18 were positively correlated with anticancer activity, clustering with the 12 most active compounds based on their IC_50_ values. Six of these variables—LogP, Csp3, b_1rotN, LogS, TPSA, and lip_don—are related to drug-likeness potential. Thus, we then ranked the 12 compounds according to these six variables and excluded those violating the drug-likeness criteria, resulting in a shortlist of nine compounds. Next, we investigated the binding affinity of these nine shortlisted compounds with the use of molecular docking towards the PIM-1 Kinase enzyme (PDB: 2OBJ), which is overexpressed in various cancer cells. Compound **6** exhibited the best docking score among the docked compounds, with a docking score of −11.77 kcal/mol, compared to −12.08 kcal/mol for the reference PIM-1 kinase inhibitor, 6-(5-bromo-2-hydroxyphenyl)-2-oxo-4-phenyl-1,2-dihydropyridine-3-carbonitrile. To discover new PIM-1 kinase inhibitors, we designed nine novel compounds featuring hybrid structures of compound **6** and the reference inhibitor. Among these, compound **31** displayed the best binding affinity, with a docking score of −13.11 kcal/mol. Additionally, we performed PubChem database mining using the structure of compound **6** and the similarity search tool, identifying 16 structurally related compounds with various reported biological properties. Among these, compound **52** exhibited the best binding affinity, with a docking score of −13.03 kcal/mol. Finally, molecular dynamics (MD) studies were conducted to confirm the stability of the protein–ligand complexes obtained from docking the studied compounds to PIM-1 kinase, validating the potential of these compounds as PIM-1 kinase inhibitors.

## 1. Introduction

Derivatives with 1,2-dihydropyridine scaffold are attracting a lot of attention owing to the readiness of their organic synthesis using several types of multi-component reactions (MCRs) and their various reported biological activities [1,2]. In the current era, where a premium is put on speed, efficacy, and diversity in the drug discovery process, large libraries of these compounds can be established efficiently in a relatively short period of time through the aid of conventional and microwave-assisted MCRs, and have become available for high-throughput biological screening [3]. Several studies have reported on the many pharmacological activities of 1,2-dihydropyridine derivatives including antimicrobial [4], cardiotonic [5], antidepressant [6], and anticancer activities [7,8].

There is much interest in the anticancer activity of these compounds due to their ability to interfere with some biological targets that are significantly associated with the pathophysiological progress of cancer, such as the PIM-1 kinase (proto-oncogene serine/threonine-protein kinase), survivin, and phosphodiesterase family of enzymes [7,9,10]. PIM-1 is an enzyme responsible for the phosphorylation, and hence the activation, of a large number of endogenous substrates that control cell cycle, proliferation, and apoptosis, suggesting that the enzyme plays a central regulatory role in cell division, survival, and differentiation processes. The oncogenic potential of PIM-1 has been demonstrated in B-cell tumors, prostate cancer, and erythroleukemia. Recent studies proved that the expression of PIM-1 is elevated in colorectal and pancreatic cancers, and the most fierce triple-negative breast cancer [9,11].

Statistical machine-learning techniques have seen widespread use in various chemistry domains [12,13,14,15,16,17]. Advances in algorithms, data availability, and computational power have contributed to the remarkable growth of this field [14,17]. By utilizing carefully curated databases and extending high-throughput methods, supervised learning approaches have successfully established connections between the chemical structure of molecules and their physical properties. Moreover, these approaches are employed in several applications in chemical synthesis, such as enabling the simulation of structural transformations at a reduced cost compared to the classic standard first-principles techniques [13,15,16,18].

From the panel of unsupervised machine learning techniques, principal component analysis (PCA) has been extensively utilized for data size reduction in large analytical chemistry datasets [19,20,21,22]. Other studies have also employed PCA to explore environmental contexts [23,24,25,26]. In this study, we aimed to couple PCA with a dataset comprising 239 different electronic, physical, and chemical properties (descriptors) of thirty 1,2-dihydropyridine derivatives. These derivatives have previously demonstrated in vitro anticancer activity against human colorectal tumor cell lines (HT-29) [27]. The objective was to identify the descriptors directly linked to the biological activity and rank the compounds based on these identified descriptors. Additionally, molecular docking was employed to assess the binding affinity of these derivatives towards a potential protein target, PIM-1 kinase. The combined results were utilized to identify the most promising candidates in terms of drug-likeness, biological activity, and binding affinity to the target protein. 

Furthermore, the compound that showed the best profile with respect to the previously mentioned analyses was utilized to propose nine novel 1,2-dihydropyridine derivatives, and was also employed in the mining of the PubChem database to identify small organic molecules with a similar chemical scaffold while retaining a reported curation effect. All the proposed and obtained compounds were subject to a second round of molecular docking to determine those with the best binding affinity towards the PIM1-Kinase. Finally, molecular dynamics simulations studies were conducted to confirm the stability of the ligand–protein complexes obtained by the top ranked compounds. 

Table 1 provides an overview of the general scaffold of the 1,2-dihydropyridine derivatives employed in this study, along with their inhibitory activity regarding tumor cell growth.

## 2. Results and Discussion

### 2.1. PCA

PCA is a statistical method that simplifies a large dataset by reducing the number of variables while keeping most of the important information. It converts the original variables into a new set of uncorrelated variables called principal components. These components are arranged so that the first few capture the most significant variations in the data.

The PCA bi-plot in Figure 1 provides a visual representation of how the different electronic, physical, and chemical properties of the thirty investigated 1,2-dihydropyridine derivatives (Table 1) relate to each other and to the principal components.

The first two principal components (PCs) explain a significant portion of the total variance, accounting for 56.79% (34.28% for PC1 and 22.51% for PC2). This means that there is a good correlation among the different molecules studied. Looking at the variables (descriptors), it is clear that they are well distributed and aligned with the first two principal components.

In terms of the individuals (the investigated molecules), distinct clusters can be observed. It is intriguing that these clusters demonstrate a strong correlation with different factors, indicating varying degrees of influence on the individuals. 

The grey cluster comprises molecules **9**, **10**, **19**, **20**, **29**, and **30**, showing a slight positive correlation along PC1 and a significant positive correlation along PC2. The blue cluster includes molecules **1**, **2**, **5**, **6**, **13**, **14**, **15**, **16**, **23**, **24**, **25**, and **26**, exhibiting a slight positive correlation along PC1 and a moderate negative correlation along PC2. The yellow cluster contains molecules **3**, **4**, **11**, **12**, **21**, and **22**, exhibiting a negative correlation along both PCs. Finally, the green cluster comprises molecules **7**, **8**, **17**, **18**, **27**, and **28**, demonstrating a pronounced negative correlation along PC1 and a slight positive correlation along PC2. These observations highlight the discrepancy among the four groups (clusters) of molecules regarding a wide range of electronic, chemical, and physical properties. This indicates that the individuals (molecules) gathered in each cluster are similar in terms of their chemical and physical properties, along with their biological activities. 

To further explore the characteristics of the drug candidates, a more focused approach was employed. Another PCA analysis was conducted, exclusively incorporating descriptors that were found from the first PCA analysis to be strongly related to the physicochemical properties and drug likeness of the candidates. Figure 2a displays the second PCA bi-plot, in which the first two principal components (PCs) explain a significant portion of the total variance, accounting for 59.91% (30.14% for PC1 and 29.77% for PC2). Notably, the variance observed in this analysis is similar to that seen when including the entire dataset (Figure 1). These findings suggest a strong correlation between the molecules and these specific descriptors. This indicates the higher efficiency of the method used in this analysis and reveals more distinct differences among the variables. Consequently, comparing different molecules becomes more reliable. 

Among the variables, ‘mr’ and ‘vol’ contributed the most to PC1, accounting for 14.46% and 13.83%, respectively (Figure 2b). ‘LogP’ and ‘TPSA’ also showed moderate contributions to PC1, accounting for 8.69% and 8.54%, respectively. For PC2, the highest contributions came from ‘MNDO_HOMO’ and ‘h_pKb’, accounting for 15.614% and 13.21%, respectively. ‘MNDO_LUMO’ and ‘h_pKa’ exhibit an average influence along PC2, contributing 10.96% and 10.15%, respectively. Interestingly, variables that showed a moderate to high correlation along PC1 demonstrated a low to no correlation along PC2 and vice versa. This suggests that the different trends observed among the molecules are independent of each other, and each set of clustered molecules is distinct and independent.

In Figure 2a, the blue cluster consists of molecules **1**, **3**, **5**, **9**, **11**, **13**, **15**, **19**, **21**, **23**, **25**, and **29**, and shows a strong positive correlation with variables such as solvation energy (E_sol), the number of hydrogen bond acceptors (a_acc), torsional energy (E_tor), and basicity (h_pKb). In contrast, the yellow cluster includes molecules **7**, **17**, and **27**, which do not exhibit a high correlation with any specific variable. The green cluster comprises molecules **4**, **8**, **12**, **18**, **22**, and **28**, demonstrating a strong positive correlation with water solubility (logS), total polar surface area (TPSA), the energy of the most stable conformer (E_str), and the number of hydrogen bond donors (lip_don). The grey cluster consists of molecules **2**, **6**, **10**, **14**, **16**, **20**, **24**, **26**, and **30**, which show a strong positive correlation with variables such as Van Der Waals energy (E_vdw), partition coefficient (logP), Van Der Waals volume (vol), the fraction of sp3-hybridized carbon atoms (Csp3), molar refractivity (mr), and the number of rotatable bonds (b_1rotN).

Remarkably, the compounds found in the grey cluster, along with compounds **8**, **18**, and **28** from the green cluster, exhibit potent anticancer activity against HT-29 colorectal tumor cells (IC50: <10 µM; IC50 for compound **24** = 10.5 µM), as discovered by Abdel-Fattah et al. [27]. These findings suggest that the variables highlighted in the grey and green clusters likely play a significant role in determining the biological activity of the compounds. ‘MNDO_HOMO’, ‘MNDO_LUMO’, and ‘h_pKa’ are positioned between the grey and green clusters, indicating their moderate influence on both clusters. ‘logS’ and ‘E’ were positioned on the negative sides of both PCs. This suggests that a lower ‘logS’ (indicating higher hydrophobicity) and a lower ‘E’ (energy) are associated with a higher probability of compounds exhibiting a potent effect against HT-29 tumor cells. 

### 2.2. Drug Likeness

The PCA provided insights into the key variables (descriptors) that influence the anticancer activity of the compounds under investigation. These variables included ‘LogP’, ‘Csp3’, and ‘b_1rotN’, which have a significant impact on the compounds in the grey cluster, and ‘LogS’, ‘TPSA’, and ‘lip_don’, which strongly influence the compounds in the green cluster. These variables are closely associated with the drug likeness potential of newly designed drug candidates, as indicated in previous studies [28,29]. The biologically active compounds in the grey cluster (consisting of nine compounds) and green cluster (consisting of three compounds), were then ranked based on the standard values of the aforementioned descriptors, as shown in Table 2. This ranking helped to identify the compounds that not only exhibit potent in vitro anticancer activity against HT-29 colorectal tumor cells, but also show promising bioavailability and pharmacokinetic properties.

According to Lipinski’s rule of five, compounds that have a molecular weight below 500, a partition coefficient (LogP) lower than 5, fewer than 5 hydrogen bond donors (lip_don), and fewer than 10 hydrogen bond acceptors (lip_acc) are more likely to exhibit good oral bioavailability [28]. Additionally, two more criteria could be considered according to Veber’s rule, which include a total polar surface area (TPSA) of less than 140 Å^2^, and a number of rotatable bonds (b_1rotN) fewer than 10 [29]. Another factor that affects drug likeness is the fraction of sp3-hybridized carbon atoms (Csp3), which is the fraction or the number of the sp3 carbon atoms out of the total carbon count. This parameter reflects the carbon saturation of molecules and characterizes the complexity of their spatial structure. Compounds with zero Csp3 are completely planar and tend to have poor absorption and bioavailability [30]. Similarly, water solubility (LogS) plays a significant role in the absorption of newly designed drugs across biological membranes. Drug candidates with a LogS value around −4 or higher are more likely to have good oral bioavailability [31].

Among the compounds listed in Table 2, compounds **10**, **20**, and **30** violated the criteria for ‘LogP’ and ‘Csp3’. Additionally, their ‘LogS’ values deviated significantly from −4 compared to the other compounds. Compounds **8**, **18**, and **28** violated only the ‘Csp3’ criterion. On the other hand, compounds **2**, **6**, **14**, **16**, **24**, and **26** did not violate any of the investigated drug-likeness criteria, suggesting that they are expected to have favorable bioavailability in addition to their previously reported potent in vitro anticancer activity against HT-29 cells.

### 2.3. Molecular Docking

The PCA and drug likeness analyses played a crucial role in guiding our selection process for compounds that were to be investigated further through molecular docking. Based on the PCA results, we specifically targeted compounds that belonged to both the grey and green clusters, resulting in a total of 12 compounds. To prioritize our focus, we ranked these 12 compounds based on drug likeness-related descriptors. As a result, only nine compounds met the criteria and were selected for subsequent investigation using molecular docking, which helped to explore the binding affinities of the shortlisted compounds to the PIM-1 kinase (PDB ID: 2OBJ). This enzyme has been identified as a significant molecular target in cancer management and treatment. The overexpression of PIM-1 kinase has been associated with various cancer-related processes, including cell proliferation, cell cycle progression, apoptosis, invasion, and glycolysis [9]. Some 1,2-dihydropyridine derivatives were reported to have an appreciable inhibitory potential against PIM-1 kinase enzyme [9]. Among them was (6-(5-bromo-2-hydroxyphenyl)-2-oxo-4-phenyl-1,2-dihydropyridine-3-carbonitrile), which exhibited an IC_50_ value of 0.05 uM and was also co-crystallized and bound to the enzyme (PDB ID: 2OBJ). In the current study, we used this compound as a reference inhibitor of PIM-1 kinase in our in silico studies, so that we could conduct effective comparisons and prioritize the newly proposed compounds for further testing. 

As depicted in Table 3, all nine docked 1,2-dihydropyridine derivatives demonstrated a favorable fit within the binding site of the PIM-1 kinase enzyme, exhibiting binding affinities comparable to that of the co-crystallized reference ligand (ref). The docking scores of the compounds ranged from −11.77 to −9.27 kcal/mol, while the co-crystallized ligand scored −12.08 kcal/mol. The compounds successfully recapitulated a significant number of the interactions observed with the co-crystallized ligand, including the arene-H hydrophobic interaction with Val52 and the H-bonding interactions with Lys67, Glu89, and Asp186. Noteworthy, the most significant interaction, as per Cheney et al., was the H-bond formed between the carbonyl group of the ligand’s pyridone ring with the Lys67 side chain in the binding site [9]. Additionally, the arene–H interaction between the Val52 side chain and the bromophenol moiety of the ligand confers extra stability to the complex. Figure 3 illustrates the 2D and 3D interactions between compound **6** and the amino acid residues within the binding site of the PIM-1 kinase.

Based on the docking results presented in Table 3, compounds **6**, **14**, and **16** exhibited the highest binding affinities, as evidenced by their lower docking scores, and exhibited profound anticancer activity against HT-29 cells as per their previously reported IC_50_ values of 0.7, 9.3, and 1.5 μM, respectively, along with appreciable drug likeness potential.

Furthermore, we utilized the chemical scaffold of compound **6** and that of the ligand inhibitor that was co-crystallized with PIM-1 kinase (PDB: 2OBJ) to design nine novel compounds (**31–39**) bearing the hybrid structural features of the previously mentioned two compounds, as shown in Table 4. 

These nine proposed designs, derived from compound **6** and the co-crystallized ligand, were shown to fulfill the drug-likeness criteria according to Lipinski’s and Veber’s rules compared to the previous candidates listed in Table 2. Upon docking compounds **31**–**39** to the PIM-1 kinase enzyme (2OBJ), they were found to fit properly in the binding site, with docking score values ranging from −13.11 to −11.73 kcal/mol (Table 5). These compounds exhibited a similar amino acid interaction pattern to that of compound **6**, indicating an appreciable PIM-1 kinase inhibitory potential. Compound **31** exhibited the best binding affinity to the target enzyme, with a docking score of −13.11 kcal/mol compared to the −12.08 kcal/mol obtained by the co-crystallized ligand inhibitor, as shown in Figure 4.

We subsequently employed the similarity search tool in the PubChem database to identify other compounds with similar chemical scaffolds to compound **6** that also possess reported curation effects. This approach allowed us to investigate the PIM-1 kinase-inhibitory potential of existing structurally similar compounds. Such an investigation highlights the importance of database mining in drug discovery, as it enables the identification of new clinical applications for these known compounds.

In this context, we applied the fingerprint Tanimoto-based two-dimensional similarity search with a similarity threshold of 90%. Upon excluding the homologues of compound **6**, sixteen compounds were finally retained (compounds **40**–**55**). Table 6 illustrates the chemical structure of these compounds along with their reported curation effect(s). 

The sixteen selected compounds were subject to molecular docking to the PIM-1 Kinase enzyme (2OBJ). Similarly, these compounds were found to fit properly in the binding site of the target enzyme, with docking score values ranging from −13.03 to −8.28 kcal/mol (Table 7), with quite similar amino acid interactions patterns to compound **6**, which indicates an appreciable PIM-1 kinase inhibitory potential as well. Compound **52** exhibited the best binding affinity to the enzyme, with a docking score of −13.03 kcal/mol compared to −12.08 kcal/mol for the co-crystallized ligand inhibitor, as shown in Figure 5.

### 2.4. Molecular Dynamics (MD)

To confirm the stability of the protein–ligand complexes obtained upon docking the studied compounds to PIM-1 kinase (2OBJ), we performed molecular dynamics studies on compounds **6** (which had the best docking score among the original thirty 1,2-dihydropyridine derivatives), 31 (which had the best docking score among the nine proposed designs), and 52 (which the best docking score among compounds with similar structures to the 1,2-dihydropyridine derivatives with curation effects), along with the co-crystallized reference ligand (ref). These compounds were identified as the top candidates based on the previous analyses. 

Molecular dynamics simulations showed that the studied compounds afforded more H-bond interactions than those retained in the molecular docking. For example, compound **6** showed only two crucial interactions with Lys67 and Val52 upon docking; however, it showed more H-bond interactions with Leu44 and Glu89 (Figure 6e). This outcome is expected, as the MD simulations provide a more dynamic and realistic picture of the protein–ligand complex over time. These interactions can change as the complex undergoes conformational changes or obtains different degrees of flexibility, offering insights into the dynamics of H-bond formation when compared to what occurs in the docking simulations, which are often static and based on a limited number of conformations.

Figure 6 shows the MD simulation analysis of the protein–compound (cpd) complexes 2OBJ-cpd6 (red), 2OBJ-cpd31 (green), 2OBJ-cpd52 (purple), and 2OBJ-ref (black). The four complexes were shown to be stable based on an analysis of the total number of H-bonds (6a), SASA (6b), RMSD (6c), and RoG (6d) in the production phase of MD simulation. The results showed that the three compounds exhibited a stable binding mode in the binding site of the PIM-1 kinase, similarly to the co-crystallized ligand (ref), and maintained their initial docking poses. The root mean square deviation (RMSD) curves indicate equilibration of the four systems after 20 ns, with average RMSD values of 2.10, 2.04, 2.23, 2.48 Å for 2OBJ-cpd6, 2OBJ-cpd31, 2OBJ-cpd52, and 2OBJ-ref complexes, respectively. The RMSD values of the 2OBJ-cpd6, 2OBJ-cpd31, 2OBJ-cpd52 complexes were slightly lower than the reference, which means that these ligands would not disrupt the stability of the protein and the function of the enzyme would not be altered. Notably, it was observed that, at 70 ns, the RMSD value of the 2OBJ-ref complex showed a slight elevation, which may be attributed to the free rotation of its unsubstituted phenyl ring (as was seen upon visual inspection of the MD trajectory). Similar results were obtained by Akintemi et al. upon studying the molecular dynamic simulations of the complexes between Sulfonylurea Receptor 1 (SUR1) and the polyphenols Galagin and Quercetin [46]: Galagin possesses an unsubstituted phenyl group and is characterized by a higher RMSD in comparison with Quercetin, which has the same structure as Galagin with a substituted phenyl group. Additionally, the numbers of the radius of gyration (RoG), the solvent-accessible surface area (SASA), and hydrogen bonds (H-bonds) average at around 19.4 Å, 14,500 Å^2^, and 80, respectively. The per-residue root mean square fluctuations (RMSFs) for the enzyme chain (6e) were plotted for the four complexes. The RMSF curves of the four complexes showed an almost identical fluctuation pattern. The highest fluctuations occurred around the Val52 and Lys67, which are the crucial residues involved in the binding pattern of the co-crystallized ligand to the PIM-1 kinase. 

Furthermore, free energy calculations (based on Van der Waals, and electrostatic interactions) for the four complexes were conducted based on the NAMD output using MolAICal version 1.3 [47,48], as follows: ΔG = G_complex_ − G_ligand_ − G_protein_

where G = E_vdw_ + E_Elec_, with solvation correction included in the E_Elec_ term as described in the MolAICal methodology [47,48].

Based on the ΔG analysis shown in Table 8, it was confirmed that the four complexes are stable, with complex 2OBJ-cpd31 being the most stable among them.

## 3. Methods

### 3.1. Computing the Descriptors of the 1,2-Dihydropyridine Derivatives

The chemical structures of the previously reported 1,2-dihydropyridine derivatives (Table 1) were drawn using the builder tool of the molecular operating environment software (Molecular Operating Environment (MOE), 2022.02 Chemical Computing Group ULC, 1010 Sherbooke St. West, Suite #910, Montreal, QC, Canada, H3A 2R7, MOE2022.v11.18.1) and compiled into one database. Then, all the 239 2D and 3D descriptors listed by MOE related to electronic, physical, and chemical properties were computed using the descriptor calculator tool of the software. 

### 3.2. Principal Component Analysis (PCA)

The PCA methodology adopted for this study is the same as that used in the work by Younes and Grasset [24]. It was performed using XLSTAT 2014. PCA is the bedrock technique for dimensionality reductions and is commonly used in different scientific and engineering fields. It was employed to reveal a lower-dimensional pattern that is most likely representative of the whole dataset. This will help create models that best interpret the scientific concept at hand. From a statistical perspective, PCA is the mathematical presentation of singular value decomposition (SVD). It will, specifically, yield a new data-driven coordinates system that represents the statistical variations in the investigated dataset. In another sense, it reveals the highest variance based on projections of the given data [19,49].

### 3.3. Molecular Docking

The target protein PIM-1 kinase was downloaded from the protein databank (PDB ID: 2OBJ). Molecular Operating Environment (MOE), 2022.02 was used to perform the docking runs and visualize the docking poses and interactions. The protein structures were prepared with the help of the Quickprep tool in MOE. The chemical structure of the co-crystallized ligand was drawn using the MOE builder tool and added to the compounds database to be docked among the compounds for the purpose of docking protocol validation. The energy of all ligands was minimized, and they were prepared using the database wash tool to adjust their formal charges. The default docking protocol adopting the triangle matcher as a placement method and London dG as a scoring function was applied. The output docking poses were explored and ranked according to their scores and the interactions they exhibited with the amino acid residues at the binding site. Self-docking of the co-crystallized ligand showed root mean square deviation (RMSD) values of less than two [50].

### 3.4. Molecular Dynamics (MD) Simulations

MD simulations of the four complexes were performed using the NAMD 2.13 software installed on the HPC platform in the American University of the Middle East (AUM), Kuwait. The calculations were performed utilizing Amber10: EHT force field [Ref.] for the protein–ligand complex and the TIP3P water model. The complex was first minimized, followed by heating to 310 K, then equilibration (at 1 atm, and 310 K) for 200 ps, and finally the production phase (for 100 ns). A 0.1 M NaCl was added to the protein–water solution [51,52]. The periodic boundary condition was utilized with a cubic simulation box. After the simulation, the trajectories were clustered and analyzed, as will be shown in the results section using VMD 1.9.3 software and in-house codes [53]. A free energy calculation (based on Van der Waals and electrostatic interactions) of the four complexes was conducted based on NAMD output files using the MolAICal protocol [47,48].

## 4. Conclusions

In conclusion, our study successfully identified key physicochemical properties correlated with the anticancer activity of thirty 1,2-dihydropyridine derivatives using Principal Component Analysis (PCA). By focusing on descriptors related to drug-likeness, we refined our dataset to nine promising candidates. Among these, compound **6** demonstrated the best binding affinity towards the PIM-1 kinase enzyme in terms of its docking score value. Further exploration led to the design of nine novel compounds, with compound **31** exhibiting the highest binding affinity. Additionally, data mining using the PubChem database revealed structurally related compounds, with compound **52** showing significant binding affinity towards the PIM-1 kinase. Molecular dynamics studies confirmed the stability of the protein–ligand complexes, reinforcing the therapeutic potential of these compounds. The complex between compound **31** and the PIM-1 kinase was shown to be the most stable in terms of ΔG values. This comprehensive approach highlights the critical role of data mining in discovering new potential anticancer drugs, underscoring the effectiveness of integrating machine learning, database mining, and molecular modeling in the drug discovery process, and paving the way for the development of innovative anticancer therapies targeting PIM-1 kinase. Moving forward, we aim to synthesize the proposed compounds and conduct further bioassays to confirm their inhibitory potential against the target enzyme.

## Figures and Tables

**Figure 1 pharmaceuticals-17-00880-f001:**
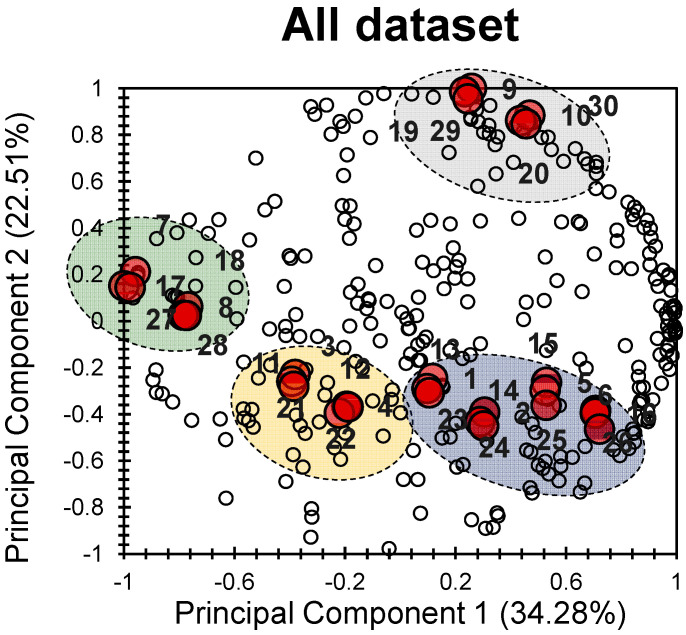
PCA bi-plot of the 30 investigated molecules (individuals) following 239 variables (descriptors); the big red bullets present the individuals and the small white bullets present the factors.

**Figure 2 pharmaceuticals-17-00880-f002:**
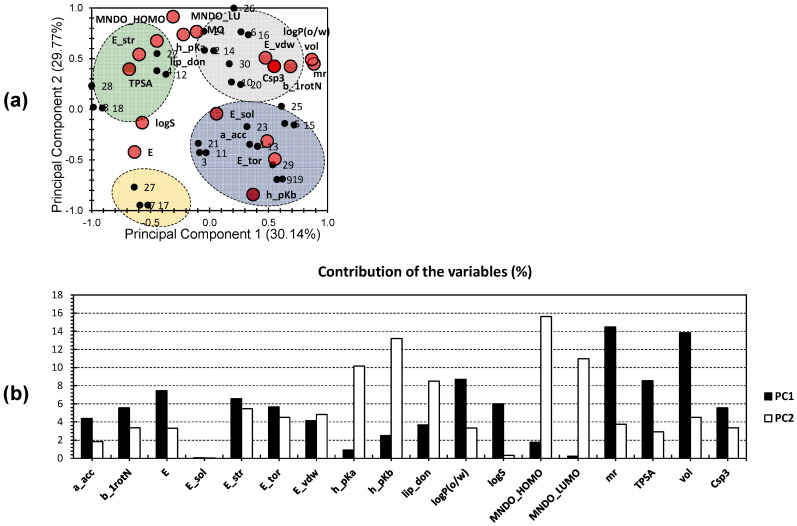
(**a**) PCA bi-plot of the 30 investigated molecules (individuals) following the 18 hand-picked variables (descriptors); the big red bullets present the variables, and the small black bullets present the individuals. (**b**) Contribution of the investigated factors toward the first two PCs.

**Figure 3 pharmaceuticals-17-00880-f003:**
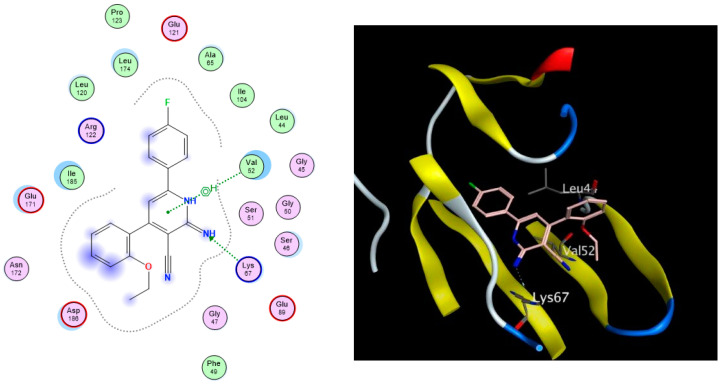
The 2D (**left**) and 3D (**right**) interactions of compound **6** with the amino acid residues in the binding site of PIM-1 kinase.

**Figure 4 pharmaceuticals-17-00880-f004:**
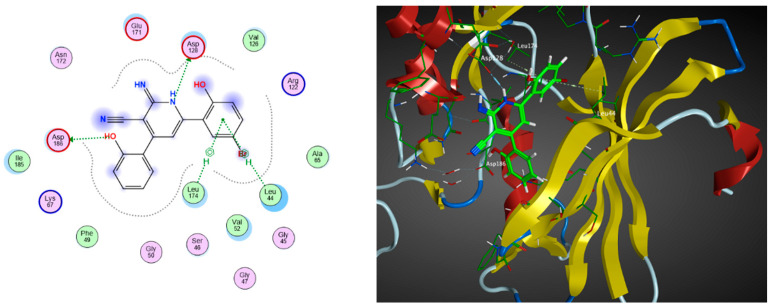
The 2D (**left**) and 3D (**right**) interactions of compound **31** with the amino acid residues in the binding site of PIM-1 kinase.

**Figure 5 pharmaceuticals-17-00880-f005:**
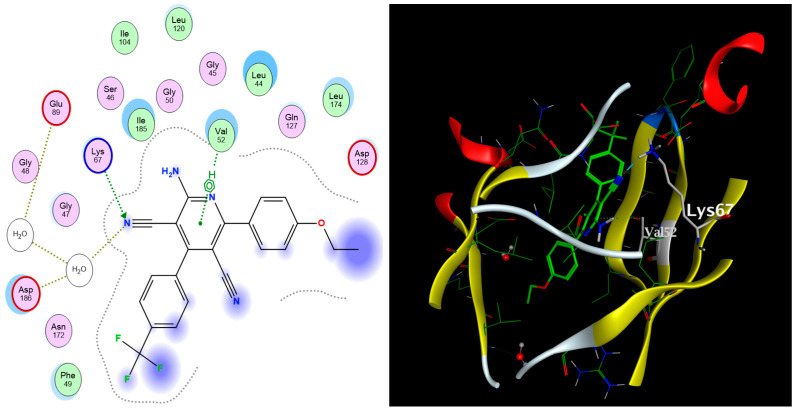
The 2D (**left**) and 3D (**right**) interactions of compound **52** with the amino acid residues in the binding site of PIM-1 kinase.

**Figure 6 pharmaceuticals-17-00880-f006:**
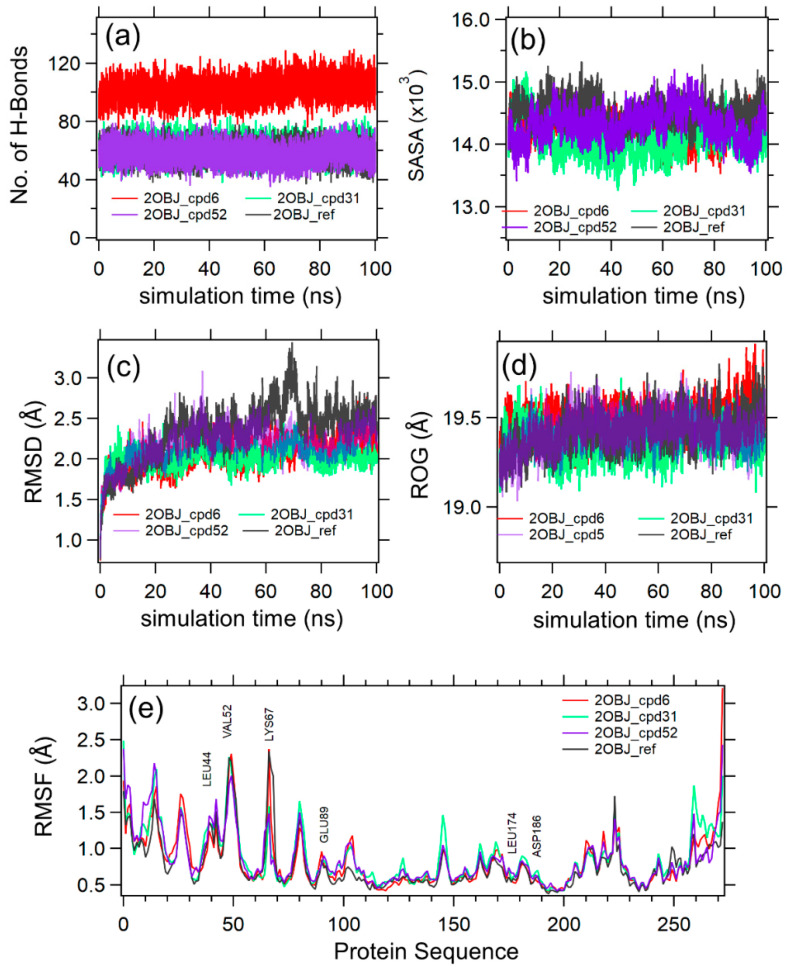
Molecular dynamics simulation data analysis for 2OBJ-cpd6, 2OBJ-cpd31, 2OBJ-cpd52, and 2OBJ-ref complexes. (**a**–**d**) show the H-bonds, SASA, RMSD, RoG and for the 2OBJ-cpd6 (red), 2OBJ-cpd31 (green), 2OBJ-cpd52 (purple), and 2OBJ-ref (black) complexes versus the simulation time in ns. (**e**) shows the per-residue RMSF from the protein sequence of 2OBJ-cpd6 (red), 2OBJ-cpd31 (green), 2OBJ-cpd52 (purple), and 2OBJ-ref (black) complexes.

**Table 1 pharmaceuticals-17-00880-t001:** The general scaffold and the chemical structures of thirty previously reported 1,2-dihydropyridine derivatives and their inhibitory tumor cell growth activity [27].

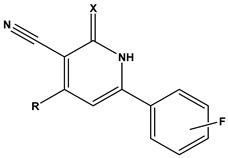									
Compound	Fluoro Position	R	X	HT-29 Growth Inhibition IC_50_ ± SD (μM)	Compound	Fluoro Position	R	X	HT-29 Growth Inhibition IC_50_ ± SD (μM)
**1**	4-F	2-methoxyphenyl	O	>50	**16**	3-F	2-ethoxyphenyl	NH	1.50 ± 0.1
**2**	4-F	2-methoxyphenyl	NH	6.30 ± 0.8	**17**	3-F	2-furanyl	O	>50
**3**	4-F	2-hydroxyphenyl	O	42.5 ± 1	**18**	3-F	2-furanyl	NH	1.26 ± 0.2
**4**	4-F	2-hydroxyphenyl	NH	12.70 ± 0.7	**19**	3-F	3,4-dichlorophenyl	O	>50
**5**	4-F	2-ethoxyphenyl	O	>50	**20**	3-F	3,4-dichlorophenyl	NH	3.96 ± 0.5
**6**	4-F	2-ethoxyphenyl	NH	0.70 ± 0.1	**21**	2-F	2-hydroxyphenyl	O	>50
**7**	4-F	2-furanyl	O	>50	**22**	2-F	2-hydroxyphenyl	NH	>50
**8**	4-F	2-furanyl	NH	3.46 ± 0.2	**23**	2-F	2-methoxyphenyl	O	>50
**9**	4-F	3,4-dichlorophenyl	O	>50	**24**	2-F	2-methoxyphenyl	NH	10.50 ± 0.4
**10**	4-F	3,4-dichlorophenyl	NH	2.18 ± 0.1	**25**	2-F	2-ethoxyphenyl	O	12.30 ± 1.0
**11**	3-F	2-hydroxyphenyl	O	>50	**26**	2-F	2-ethoxyphenyl	NH	2.50 ± 0.3
**12**	3-F	2-hydroxyphenyl	NH	>50	**27**	2-F	2-furanyl	O	>50
**13**	3-F	2-methoxyphenyl	O	12.70 ± 1.2	**28**	2-F	2-furanyl	NH	8.82
**14**	3-F	2-methoxyphenyl	NH	9.30 ± 0.8	**29**	2-F	3,4-dichlorophenyl	O	>50
**15**	3-F	2-ethoxyphenyl	O	10.20 ± 1.2	**30**	2-F	3,4-dichlorophenyl	NH	5.74 ± 0.4

**Table 2 pharmaceuticals-17-00880-t002:** Drug-likeness data of the biologically active compounds gathered in the grey and green clusters.

Compound		Drug Likeness-Related Descriptors				
	LogP	Csp3	b_1rotN	LogS	TPSA	lip_don
**2**	4.095	0.05	3	−5.467	68.9	2
**6**	4.436	0.1	4	−5.794	68.9	2
**8**	2.856	0	2	−5.168	72.81	2
**10**	5.360	0	2	−6.885	59.67	2
**14**	4.132	0.05	3	−5.467	68.9	2
**16**	4.473	0.1	4	−5.794	68.9	2
**18**	2.893	0	2	−5.168	72.81	2
**20**	5.397	0	2	−6.885	59.67	2
**24**	4.093	0.05	3	−5.467	68.9	2
**26**	4.434	0.1	4	−5.794	68.9	2
**28**	2.854	0	2	−5.168	72.81	2
**30**	5.358	0	2	−6.885	59.67	2

**Table 3 pharmaceuticals-17-00880-t003:** Docking scores and amino acids interactions obtained upon docking the shortlisted 1,2-dihydropyridine derivatives to the PIM-1 kinase enzyme.

Compound	Docking Score (kcal/mol)	Types of Interactions
Co-crystallized ligand	−12.08	Val52 (arene–H) Lys67 (H-bond) Ile185 (arene–H) Glu89 (H-bond) Asp186 (H-bond)
**6**	−11.77	Val52 (arene–H) Lys67 (H-bond)
**14**	−11.33	Leu44 (arene–H) Leu174 (arene–H) Glu89 (H-bond) Asp186 (H-bond)
**16**	−11.26	Val52 (arene–H) Lys67 (H-bond) Leu174 (arene–H)
**24**	−11.06	Val52 (H-bond) Lys67 (H-bond) Leu174 (arene–H)
**2**	−10.99	Val52 (arene–H) Lys67 (H-bond) Leu174 (arene–H)
**26**	−10.41	Leu44 (arene–H) Val52 (H-bond) Lys67 (H-bond) Leu174 (arene–H)
**18**	−9.83	Leu44 (arene–H) Val52 (H-bond) Lys67 (H-bond)
**28**	−9.44	Leu44 (arene–H) Val52 (arene–H) Lys67 (H-bond)
**8**	−9.27	Leu44 (arene–H) Val52 (H-bond) Lys67 (H-bond)

**Table 4 pharmaceuticals-17-00880-t004:** The general scaffold of the newly designed 1,2-dihydropyridine derivatives.

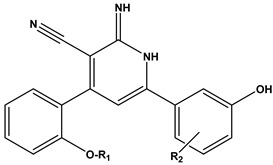		
Compound	R1	R2
**31**	H	5-Br
**32**	CH_3_	5-Br
**33**	CH_2_CH_3_	5-Br
**34**	H	5-F
**35**	CH_3_	5-F
**36**	CH_2_CH_3_	5-F
**37**	H	4-F
**38**	CH_3_	4-F
**39**	CH_2_CH_3_	4-F

**Table 5 pharmaceuticals-17-00880-t005:** Docking scores and amino acid interactions obtained upon docking the proposed 1,2-dihydropyridine designs (**31**–**39**) to the PIM-1 kinase enzyme.

Compound	Docking Score (kcal/mol)	Types of Interactions
**31**	−13.11	Leu44 (arene–H) Asp128 (H-bond) Leu174 (H-bond) Asp186 (H-bond)
**32**	−11.97	Val52 (arene–H) Asp128 (H-bond) Lys67 (H-bond)
**33**	−12.75	VAl52 (arene–H) Phe49 (H-bond)
**34**	−11.79	Val52 (arene–H) Lys67 (H-bond) Asp128 (H-bond) Asn172 (H-bond)
**35**	−12.43	Val52 (arene–H) Lys67 (H-bond) Asp128 (H-bond) Leu174 (arene–H)
**36**	−11.73	Leu44 (H-bond) Val52 (arene–H) Lys67 (H-bond)
**37**	−12.17	Val52 (arene–H) Lys67 (H-bond) Asp128 (H-bond) Leu174 (arene–H)
**38**	−12.05	Val52 (arene–H) Lys67 (H-bond) Asp128 (H-bond) Leu174 (arene–H)
**39**	−12.34	Val52 (arene–H) Lys67 (H-bond) Asp128 (H-bond) Leu174 (arene–H)

**Table 6 pharmaceuticals-17-00880-t006:** The chemical structures of compounds **40–55** and their reported curation effect(s).

Compound	Structure	Curation Effect	Reference
**40**	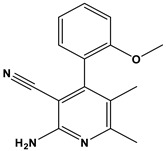	Mitogen-activated protein Kinase-activated protein Kinase-2 inhibitors.	[32]
**41**	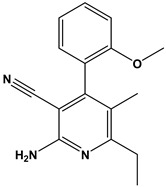	Inhibitors of Lassa virus’ entry into cells.	[33]
**42**	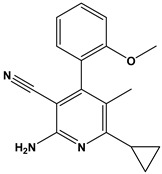	Mitogen-activated protein Kinase-activated protein Kinase-2 inhibitors.	[34]
**43**	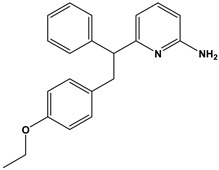	Beta-secretase inhibitors for the treatment of Alzheimer’s disease.	[35]
**44**	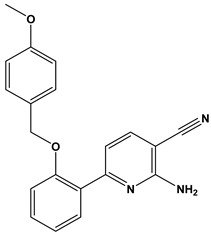	Treatment of diseases associated with NF-kB activity, in particular for the treatment of inflammatory diseases.	[36]
**45**	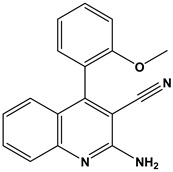	Treatment and/or prophylaxis of diseases which are associated with DPP IV, such as diabetes, particularly non-insulin-dependent diabetes mellitus, and impaired glucose tolerance.	[37]
**46**	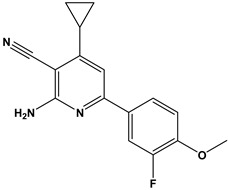	Inhibitor of nuclear factor kappa-B kinase subunit beta, and inhibitor of Mycobacterium tuberculosis growth.	[38]
**47**	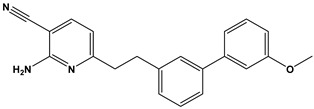	Treatment or prophylaxis of cognitive impairment, Alzheimer’s disease, neurodegeneration, and dementia.	[39]
**48**	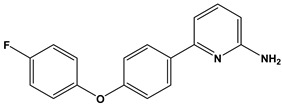	Sodium channel blockers.	[40]
**49**	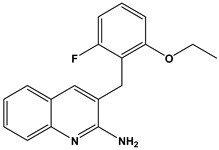	Treating neuro-degenerative and neuropsychiatric disorders.	[41]
**50**	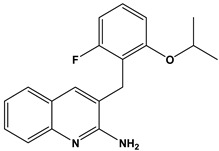	Treating neuro-degenerative and neuropsychiatric disorders.	[41]
**51**	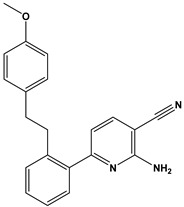	Anti-inflammatory activity.	[42]
**52**	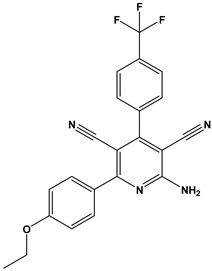	Inhibitors of sodium–calcium exchange to be used for the prevention and/or management of cardiovascular diseases.	[43]
**53**	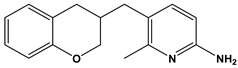	Inhibit the growth of Mycobacterium tuberculosis.	[38]
**54**	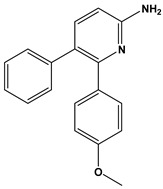	Treatment of cardiovascular and cardio-metabolic pathologies.	[44]
**55**	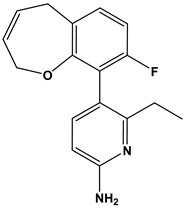	Neuropeptide FF receptor antagonists	[45]

**Table 7 pharmaceuticals-17-00880-t007:** Docking scores and amino acids interactions obtained upon the docking of compounds (**40**–**55**) to the PIM-1 kinase enzyme.

Compound	Docking Score (kcal/mol)	Types of Interactions
**40**	−10.33	Val52 (arene–H) Lys 67 (H-bond)
**41**	−10.69	Val52 (arene–H) Lys 67 (H-bond)
**42**	−8.634	Val52 (arene–H) Lys 67 (H-bond)
**43**	−9.87	Val52 (arene–H) Asp128 (H-bond)
**44**	−10.47	Leu44 (arene–H) Val52 (arene–H) Lys 67 (H-bond)
**45**	−9.45	Val52 (arene–H) Ile185 (arene–H)
**46**	−9.51	Val52 (arene–H) Lys67 (H-bond) Leu174 (arene–H)
**47**	−11.62	Lys67 (H-bond)
**48**	−9.73	Leu174 (arene–H) Asp186 (H-bond)
**49**	−9.64	Val52 (arene–H)
**50**	−9.82	Leu174 (arene–H) Ile185 (arene–H)
**51**	−11.03	Val52 (arene–H)
**52**	−13.03	Val52 (arene–H) Lys67 (H-bond) Leu174 (arene–H)
**53**	−8.28	Val52 (arene–H) Leu174 (arene–H)
**54**	−9.38	Val52 (arene–H) Asp128 (H-bond) Ile185 (arene–H)
**55**	−9.06	Val52 (arene–H)

**Table 8 pharmaceuticals-17-00880-t008:** ΔG values of the four PIM-1 kinase–ligand complexes.

Complex Name	Complex E(vdW)	Complex E(Elec) ^1^	Protein E(vdW)	Protein E(Elec) ^1^	Ligand E(vdW)	Ligand E(Elec) ^1^	ΔG (kcal/mol)
2OBJ-cpd31	−1309.626	−11,259.7506	−1269.751	−11,125.7145	9.8251	−119.6795	−64.06
2OBJ-cpd6	−1298.2813	−11,220.477	−1272.5471	−11,127.3604	10.6892	−106.0379	−23.5
2OBJ-ref	−1274.8976	−11,169.5821	−1280.6223	−11,071.7319	12.1326	−98.3834	−5.87
2OBJ-cpd52	−1285.4504	−11,176.3037	−1269.4016	−11,088.5128	11.7883	−100.4193	−15.21

^1^ Solvation correction included in EElec term as described in the MolAICal methodology [47,48].

## Data Availability

Data is contained within the article.

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
