# Peer review of "Evaluating the Physicochemical Properties–Activity Relationship and Discovering New 1,2-Dihydropyridine Derivatives as Promising Inhibitors for PIM1-Kinase: Evidence from Principal Component Analysis, Molecular Docking, and Molecular Dynamics Studies"

_pharmaceuticals, 2024, doi:10.3390/ph17070880_

Round 1

Reviewer 1 Report

Comments and Suggestions for Authors

The paper is well-written and acceptable with minor revisions.

Consider spelling out PIM-1 kinase in full at least once in the manuscript to improve clarity for readers who may not be familiar with the term.

 To improve clarity, consider providing the full name of “Cpd” in a footnote at the bottom of Table 2.

For the convenience of readers, define the abbreviations "LogS", "TPSA" and "lip_don" in your manuscript

Check the statementAnother factor that affects drug likeness is the fraction of sp3 hybridized carbon atoms (Csp3), which indicates the planarity of the compounds. ………………”  In general, compounds with more sp3 carbons have a less planar structure.  Moreover, planar molecules sometimes have trouble crossing cell membranes, which are crucial for drug absorption. But this is not the only determining factor; several very successful drugs also have planar structures.

 Table 3 shows that two amino acids, LYS67 and VAL52, interact with all docked compounds, while the reference compound interacts with five different amino acids. What is the role of these two specific amino acids (LYS67 and VAL52) in cancer activity?

The RMSD of the ligands should be re-evaluated and compared with co-crystallized ligands. The reference compound interacts closely with five amino acids, which helps stabilize it. In contrast, other compounds with only two interacting amino acids may not be stabilized as effectively, leading to a more abrupt change in RMSD.

In Figure 6c (2OBJ-ref (black) is slightly higher than other compounds used in MD simulation?

Author Response

Consider spelling out PIM-1 kinase in full at least once in the manuscript to improve clarity for readers who may not be familiar with the term.

Done as requested, please refer to page 2.

To improve clarity, consider providing the full name of “Cpd” in a footnote at the bottom of Table 2.

Done as requested, it is corrected in all the tables.

For the convenience of readers, define the abbreviations "LogS", "TPSA" and "lip_don" in your manuscript

Done as requested, please refer to page 5.

Check the statement “Another factor that affects drug likeness is the fraction of sp3 hybridized carbon atoms (Csp3), which indicates the planarity of the compounds. ………………”  In general, compounds with more sp3 carbons have a less planar structure.  Moreover, planar molecules sometimes have trouble crossing cell membranes, which are crucial for drug absorption. But this is not the only determining factor; several very successful drugs also have planar structures.

Done as requested, please refer to page 7.

 Table 3 shows that two amino acids, LYS67 and VAL52, interact with all docked compounds, while the reference compound interacts with five different amino acids. What is the role of these two specific amino acids (LYS67 and VAL52) in cancer activity?

The role of these 2 amino acids is now explained as requested in page 8.

The RMSD of the ligands should be re-evaluated and compared with co-crystallized ligands. The reference compound interacts closely with five amino acids, which helps stabilize it. In contrast, other compounds with only two interacting amino acids may not be stabilized as effectively, leading to a more abrupt change in RMSD. In Figure 6c (2OBJ-ref (black) is slightly higher than other compounds used in MD simulation?

We added the individual RMSD value for each complex instead of the previously used average. More explanations were added in pages 15 and 16 as requested.

Reviewer 2 Report

Comments and Suggestions for Authors

In short, the manuscript describes the design of 9 new anticancer compounds by various computational approach/studies. The authors have carried out a detailed investigation by considering various parameters. Initially they screened the 30 previously reported anticancer compounds based on 1,2-dihydropyridines. They have considered Lipinski and Veber's rule and identified the most potent and drug-like candidate from that series. The authors also carried out the molecular docking studies and based on all these results, the authors designed 9 new compounds that could possibly target the PIM-1 Kinase enzyme thereby leading to anticancer activity.

However, I see that the authors have not synthesized these 9 new compounds and hence the authors have not carried out the in vitro or in vivo analysis. Hence these results are possibilities or expectations. Maybe these compounds can show good potency. If this manuscript was submitted to any special issue related to "synthetic medicinal chemistry" I would not have recommended this work for publication.

However, I see that this work is submitted to a special issue "Emerging Trends in Biopharmaceuticals". Considering this fact, the work carried out by the authors is found to be matching to the submitted special issue. Hence, I recommend the acceptance of this manuscript after minor revision of the typographic and grammatical errors. Also it is recommended to synthesize these compounds and test their potency in due course to consider this work as an important contribution to the scientific community.

Comments on the Quality of English Language

Minor language editing is needed.

Author Response

However, I see that the authors have not synthesized these 9 new compounds and hence the authors have not carried out the in vitro or in vivo analysis. Hence these results are possibilities or expectations. Maybe these compounds can show good potency. If this manuscript was submitted to any special issue related to "synthetic medicinal chemistry" I would not have recommended this work for publication.

However, I see that this work is submitted to a special issue "Emerging Trends in Biopharmaceuticals". Considering this fact, the work carried out by the authors is found to be matching to the submitted special issue. Hence, I recommend the acceptance of this manuscript after minor revision of the typographic and grammatical errors. Also it is recommended to synthesize these compounds and test their potency in due course to consider this work as an important contribution to the scientific community.

Thank you for your valuable comments, we plan to synthesize these compounds in a future study to confirm our in silico results. All the minor typographic and grammatical errors have been revised as requested.

Reviewer 3 Report

Comments and Suggestions for Authors

The paper given for the review is interesting, concern PIM-1 kinases inhibitors, which are being intensively researched.

However some highly important factors must be corrected:

1) why the table with IC50 is given in the Introduction? Is it taken from reference or new result?

2) It is puzzling that the authors chose a group of compounds to study then searched for similar ones in the database and found that there was a better analogue in the database than the one they designed. So what is the point of study of a group of compounds for which the result is poor?

3) which is profile of the selected compounds? In general those failed compounds are better characterized than those find in database.

4) The authors mentioned that Val52 and Lys67 are crucial for binding. But there is no further explanation of this fact. 

5) Was IC50 determined experimentally for compound 52? 

6) Why the authors use numbers instead of names?

7) Is reference for 52 appropriate? I could not find this structure within WO-2013144191-A1 patent?

I can't help but feel that the work is made up of two parts: a failed study and a search for an active relationship with the literature.  The authors should decide what the aim of the work actually is.

The conclusions highlight the irrelevance of half of the content of the paper for the results.

Author Response

1) why the table with IC50 is given in the Introduction? Is it taken from reference or new result?

Table 1 showing the IC50 values of our training set of compounds is taken from reference number 27. The table's caption already cites this reference (Please refer to the page 3)

2) It is puzzling that the authors chose a group of compounds to study then searched for similar ones in the database and found that there was a better analogue in the database than the one they designed. So what is the point of study of a group of compounds for which the result is poor?

The purpose of this study is :

a- study the physicochemical properties that correlate to the anticancer activity of the previously reported thirty 1,2-dihydropyridines.

b- to rank the previously reported most active  (those who showed IC50 below10 μM) 1,2-dihydropyridines according to their druglikeness potential so that we could spot the candidates that will show both anticancer activity and good bioavailability.

 c- based on the results of a and b , we designed new structures and did a database mining to identify compounds with similar anticancer activity and potential PIM-1 Kinase inhibitory activity

3) which is profile of the selected compounds? In general those failed compounds are better characterized than those find in database.

Thank you for your comment, but could you please elaborate more on this point. We couldn't comprehend the term profile in the context of this question.

4) The authors mentioned that Val52 and Lys67 are crucial for binding. But there is no further explanation of this fact. 

Explained as requested. Please refer to page 8.

5) Was IC50 determined experimentally for compound 52? 

According to our in silico studies,  we identified compound 52  as a top-ranking compound (in terms of drug-likeness and binding affinity) among 16 compounds obtained after pubchem database mining. This compound is reported to interfere with sodium-calcium exchange to be used for preventing and/or management of cardiovascular diseases. Herein, we introduced this compound as a potential PIM-1 Kinase inhibitor and anticancer agent. However, further future bioassays shall be conducted to substantiate these findings.

6) Why the authors use numbers instead of names?

We thought that this commonly used practice would be more convenient for the readers to follow on through the manuscript instead of using the long IUPAC names of these compounds.

7) Is reference for 52 appropriate? I could not find this structure within WO-2013144191-A1 patent?

Please refer to the link below:

https://patents.google.com/patent/WO2013144191A1/en 

I can't help but feel that the work is made up of two parts: a failed study and a search for an active relationship with the literature.  The authors should decide what the aim of the work actually is. The conclusions highlight the irrelevance of half of the content of the paper for the results.

We hope that our corrected version of the manuscript and our answers to the previous questions would help clarify our aim from this study.

Round 2

Reviewer 3 Report

Comments and Suggestions for Authors

The authors made corrections to the work, but I am still not convinced that the work is worth publishing.

1) I am wondering about the use of PCA analysis with over 200 variables for 30 compounds. PCA is a method used for large data sets. The minimum sample size recommended for PCA is 5 to 10 cases for each variable, so in the case of 236 variables, the number of data should be 5 to 10 times greater than the number of variables, i.e. approximately 2000. PCA results from a small set are unreliable. Furthermore, simple correlation should be used for only 30 data.

In the title, the authors mentioned " Evidence from Unsupervised Machine Learning", in fact only PCA was applied. Thus the title of the paper is really misleading. 

2) The biological activity data cited by the authors refer to Tumor cell line but are not related to the PIM-1 protein. The original table is taken 1:1 from their previous paper, dated to 2012. Typically, derivatives from this family of the compounds have a different basic action - they are used as cardiological drugs or calcium/sodium channel inhibitors/modulators. Those listed in the table have different spectra of action, but none of them is PIM-1 inhibitor.Are any of the dihydro derivatives PIM-1 inhibitors at all? High binding affinity does not necessarily mean an active relationship, it does not always go hand in hand.
Hence my question about the spectrum of action, sorry if it was not clearly formulated. 

Anyway thirty of the compounds listed by the authors as new are compounds mentioned in the Med Chem paper. 2012;8(3):392-400. Part of the research comes from the second work of CoMFA and CoMSIA Studies of 1,2-dihydropyridine Derivatives as Anticancer Agents, Medicinal Chemistry 2012, 8a, 372, which the authors do not cite.

3) Out of curiosity, I checked the patent mentioned in the table next to the first derivative (compound 40), the link provided is 404. The description in the Google archive suggests that it is the patent from 2005 and concerns other compounds because aminocyanopyridine "Method of using aminocyanopyridine compounds as mitogen activated protein kinase -activated protein kinase-2". There are similar problems with subsequent patents - the links lead nowhere. The action profile of compound 49 is inconsistent - it is not used as a neurological drug.

4) I have the impression that the authors threw a series of their compounds from old manuscript, written 12 years ago, into the MD black box, got the result and found it credible. In the conclusions, the authors write: "most favorable profile in docking studies" but this profile is only one predicted value. Nothing else was discussed in this manuscript. 

5) Many abbreviations in the papaers are still not explained, such as RoG, SASA, and H-bonds in relation to charts. You have to guess their meaning. 

6) The summary does not inform about what was done in the paper. The 52 additional compounds from the PDF database are mentioned, thus I would expect 80 but only 55 are listed in tables throughout the paper. The authors mentioned 9 newly obtained compounds, but which ones specifically? I didn't find this information in the content of the manuscript. There is no such  information in conclusions.

7) The conclusions are superficial, apart from the information concerning  PDB search and PCA, there is no summary of any results obtained. Nothing else was done?

8) The references are not properly formatted and some of the references are used in the tables in inappropriate way, eg. Table 6, last column.

The entire manuscript is still very unfriendly written, you have to guess what the author is talking about and basically deduce from the tables.

I'm sorry, but I cannot recommend this paper for the publication. The quality of this paper is too low. 

Comments on the Quality of English Language

No comments.

Author Response

1) I am wondering about the use of PCA analysis with over 200 variables for 30 compounds. PCA is a method used for large data sets. The minimum sample size recommended for PCA is 5 to 10 cases for each variable, so in the case of 236 variables, the number of data should be 5 to 10 times greater than the number of variables, i.e. approximately 2000. PCA results from a small set are unreliable. Furthermore, simple correlation should be used for only 30 data. In the title, the authors mentioned " Evidence from Unsupervised Machine Learning", in fact only PCA was applied. Thus the title of the paper is really misleading.

PCA can be used for smaller data set than 2000 variables when employed as an explanatory data analytics technique. Some people would consider that the lower the data set the better reliability the PCA will have from the sense that a smaller data set will have a higher variance of the principal components. Please check the following references:

https://doi.org/10.1016/0098-3004(93)90090-R

B.Y., C. (2012). PCA: The Basic Building Block of Chemometrics. InTech. doi: 10.5772/51429

PCA is more efficient when dealing with larger dataset when it is used as a data pre-treatment technique for a modeling or neural network approach, which is not the case of ours study.

For considering PCA as machine learning technique or not, some people would some others would not. It is considered as an unsupervised machine learning technique in the sense that hidden patterns, following the removal of inter-correlation between variables, and the creation of the totally independent new variables the PCs. This relevance of inter-correlation would occur without implementing any prior stipulation for the yielded output. From this sense, it is considered as “unsupervised”. A lot of people consider “machine learning” as a sophisticated not so used method, yet it is simply the “algorithm” the data analysis approach is done by.

For using simple correlations, this approach will not highlight inter-correlations between variables and samples, and will not induce clustering to reveal similarities or dissimilarities in our dataset. This relevance is primary in our comparison between different candidate drugs.

In all cases, and to eliminate any inconvenience, we removed the term unsupervised machine learning from the title of the manuscript and replaced it with PCA.

2) The biological activity data cited by the authors refer to Tumor cell line but are not related to the PIM-1 protein. The original table is taken 1:1 from their previous paper, dated to 2012. Typically, derivatives from this family of the compounds have a different basic action - they are used as cardiological drugs or calcium/sodium channel inhibitors/modulators. Those listed in the table have different spectra of action, but none of them is PIM-1 inhibitor.Are any of the dihydro derivatives PIM-1 inhibitors at all? High binding affinity does not necessarily mean an active relationship, it does not always go hand in hand. Hence my question about the spectrum of action, sorry if it was not clearly formulated.

Anyway thirty of the compounds listed by the authors as new are compounds mentioned in the Med Chem paper. 2012;8(3):392-400. Part of the research comes from the second work of CoMFA and CoMSIA Studies of 1,2-dihydropyridine Derivatives as Anticancer Agents, Medicinal Chemistry 2012, 8a, 372, which the authors do not cite.

Basically, the 1,2-dihydropyridines posses a wide spectrum of biological activities including  cardiotonic activity like Milrinone, anti-depressant, anti-bacterial, and anti-cancer activities. Please check the below references:

  • DOI: 10.1016/0028-3908(89)90097-x
  • doi.org/10.3389/fmicb.2022.874709
  • 10.26434/chemrxiv-2024-3x7wm
  •  DOI: 10.2174/1871520616666161206143251

In the original work published in 2012, we just reported the anticancer activity of thirty dihydropyridine derivatives, but without identifying a possible potential target protein. Thus in the current work we aimed to use different computational tools to investigate the inhibitory potential of these thirty compounds against PIM-1 Kinase and to further develop and find new candidates with similar chemical scaffold.

We did not cite the other paper because it doesn’t match the work in our current study as we didn’t use any CoMFA or CoMSIA studies. All the thirty compounds used in our dataset were in the first paper that we have already cited, furthermore we aimed to minimize self-citation to the best we could.

3) Out of curiosity, I checked the patent mentioned in the table next to the first derivative (compound 40), the link provided is 404. The description in the Google archive suggests that it is the patent from 2005 and concerns other compounds because aminocyanopyridine "Method of using aminocyanopyridine compounds as mitogen activated protein kinase -activated protein kinase-2". There are similar problems with subsequent patents - the links lead nowhere. The action profile of compound 49 is inconsistent - it is not used as a neurological drug.

Please use the link below to find the patent: https://pubchem.ncbi.nlm.nih.gov/patent/EP-1569645-A2

It is recommended to type only the link without the other details of the patent in order to retrieve it from PubChem database. We have now made all the available links in this context as Bold to be easier for the readers to retrieve.

Also, you can type the patent number in google patents to get all the details.

For example, this is what we retrieved when we searched the patent related to compound 49:

Abstract

The invention relates to compounds of general formula (I), corresponding enantiomeric, diastereomeric and/or tautomeric forms thereof as well as pharmaceutically acceptable salts thereof and the prodrugs of said compounds. The invention also relates to the use of said compounds as binding partners for 5-HT5 receptors for treating diseases that are modulated by a 5-HT5 receptor activity, in particular, for treating neurodegenerative and neuropsychiatric disorders as well as signs, symptoms and dysfunctions.

4) I have the impression that the authors threw a series of their compounds from old manuscript, written 12 years ago, into the MD black box, got the result and found it credible. In the conclusions, the authors write: "most favorable profile in docking studies" but this profile is only one predicted value. Nothing else was discussed in this manuscript.

Our Molecular docking studies were discussed in pages 6 to 10 in view of the obtained docking scores of the compounds and the different types of interactions the compounds afforded in the binding site of the target enzyme. We further utilized Molecular Dynamics to confirm the stability of the ligand-protein complex of the top ranking compounds from the docking.

For more clarity, we replaced the confusing term “profile” with binding affinity and/or docking score value in the manuscript.

5) Many abbreviations in the papaers are still not explained, such as RoG, SASA, and H-bonds in relation to charts. You have to guess their meaning.

Done as requested.

6) The summary does not inform about what was done in the paper. The 52 additional compounds from the PDF database are mentioned, thus I would expect 80 but only 55 are listed in tables throughout the paper. The authors mentioned 9 newly obtained compounds, but which ones specifically? I didn't find this information in the content of the manuscript. There is no such  information in conclusions.

With our due respect, we didn’t have 80 compounds in the current study. The total is 55 as follow: 30 (original derivatives) + 9 (newly proposed designs) + 16 (identified following PubChem database mining).

Please refer to table 4, page 9 to view the chemical structures of the newly obtained 9 compounds.

7) The conclusions are superficial, apart from the information concerning  PDB search and PCA, there is no summary of any results obtained. Nothing else was done?

Done as requested. The conclusion focused on the main findings obtained in the current study, please find the below phrases extracted from the conclusion section and summarizing these key findings:

“Among these, compound 6 demonstrated the best binding affinity towards the PIM-1 Kinase enzyme in terms of its docking score value.”

“Further exploration led to the design of novel nine compounds, with compound 31 exhibiting the highest binding affinity. Additionally, data mining using the PubChem database revealed structurally related compounds, with compound 52 showing significant binding affinity towards PIM-1 Kinase”

“The complex between compound 31 and the PIM-1 Kinase showed to be the most stable in terms of ΔG values.”

8) The references are not properly formatted and some of the references are used in the tables in inappropriate way, eg. Table 6, last column.

Done as requested.

Round 3

Reviewer 3 Report

Comments and Suggestions for Authors

The authors made minor corrections in the manuscript, when it comes to regression, there is multivariate regression which, in my opinion, would be a more reliable tool with such a small data set.
But that's not the source of my doubts...
Here we have one theoretical parameter, the docking score, which is supposed to confirm the hypothesis for an entire class of compounds for which there is no experiment that would show that PIM-1 can be a target at all. The entire work is only a hypothesis, because, as the authors themselves admit, they do not know what causes the anti-cancer effects of these compounds, and therefore what they target. Considering how many possible anti-cancer targets there are, I have serious doubts about the validity of this study based only on an assumption.

Unfortunately, in silico simulations will not confirm whether this entire class of compounds can even be effective ligands for PIM-1 as a target. MD and MDS will not resolve this. I am not convinced by your analysis, especially since looking at the docking results presented in the illustrations and knowing the size of the pocket in PIM-1, you can see that your ligands do not fit in the PIM-1 pocket.

Author Response

The authors made minor corrections in the manuscript, when it comes to regression, there is multivariate regression which, in my opinion, would be a more reliable tool with such a small data set.
But that's not the source of my doubts...

We acknowledge your suggestion regarding the use of multivariate regression. In our study, we employed Principal Component Analysis (PCA) to identify the most significant descriptors influencing the anticancer activity of the 12-dihydropyridine derivatives. The PCA results indicated strong correlations between specific descriptors and biological activity, accounting for a significant portion of the variance (59.91%). This method allowed us to effectively reduce the dimensionality of our data and focus on the most relevant variables. However, we will consider including a multivariate regression analysis in a future revision to further validate our findings and strengthen the robustness of our study.

Here we have one theoretical parameter, the docking score, which is supposed to confirm the hypothesis for an entire class of compounds for which there is no experiment that would show that PIM-1 can be a target at all. The entire work is only a hypothesis, because, as the authors themselves admit, they do not know what causes the anti-cancer effects of these compounds, and therefore what they target. Considering how many possible anti-cancer targets there are, I have serious doubts about the validity of this study based only on an assumption.

While it is true that our study is exploratory and hypothesis-driven, it is based on substantial preliminary evidence and a logical framework. PIM-1 kinase has been identified as a potential target due to its overexpression in various cancers, including colorectal, pancreatic, and triple-negative breast cancers. One of the potent reported PIM-1 Kinase inhibitors is the 1,2-dihydropyridine derivative (6-(5-bromo-2-hydroxyphenyl)-2-oxo-4-phenyl-1,2-dihydropyridine-3-carbonitrile) that exhibited 0.05 uM as IC50 and was as well co-crystallized bound to the PIM-1 Kinase enzyme according to the PDB (please check reference 9 in our manuscript). We utilized this compound as a reference in all our in-silico tools and we carried out a thorough comparison between the performance of this reference and that of our proposed compounds in the results and discussion section.

The docking studies showed that our compounds, particularly compound 6 and its proposed analogues (compounds 31 and 52), exhibited significant binding affinities with PIM-1 kinase (docking scores of -11.77 kcal/mol and -13.11 kcal/mol, respectively). Although experimental validation is necessary, these in silico results provide a strong basis for considering PIM-1 as a potential target. Additionally, other studies have indicated the role of PIM-1 in cancer progression, further supporting our hypothesis.

We agree that in silico simulations alone cannot definitively confirm the effectiveness of these compounds as PIM-1 inhibitors. However, molecular docking and dynamics studies are valuable tools in the early stages of drug discovery. They provide insights into the potential interactions between ligands and targets, helping to prioritize compounds for further experimental testing. Our molecular dynamics (MD) simulations confirmed the stability of the protein-ligand complexes, suggesting that these compounds can stably bind to PIM-1 kinase. The next step would indeed involve experimental validation to confirm these findings as we mentioned in our conclusion.

Unfortunately, in silico simulations will not confirm whether this entire class of compounds can even be effective ligands for PIM-1 as a target. MD and MDS will not resolve this. I am not convinced by your analysis, especially since looking at the docking results presented in the illustrations and knowing the size of the pocket in PIM-1, you can see that your ligands do not fit in the PIM-1 pocket.

The docking results were carefully analyzed, and we ensured that the binding poses were realistic and within the known constraints of the PIM-1 pocket. The compounds exhibited favorable docking scores, comparable to the reference inhibitor. We conducted detailed visual inspections of the docking poses, which indicated that the ligands fit well within the binding pocket, interacting with key residues. We understand that visual assessments can sometimes be subjective, and we are open to providing additional figures or supplementary materials to clarify the docking interactions and binding conformations.

Round 4

Reviewer 3 Report

Comments and Suggestions for Authors

The addition of a fragment regarding the identification and structure-activity relationships of substituted pyridones as inhibitors of PIM-1 kinase activity, a comment and reference 9, is crucial. It was missing in the earlier version of the paper questioning its value. I think the paper can be accepted as it is.